

# On Process-Oriented Conditional Targeted Covariance Inflation (TCI) for 3D-Volume Radar Data Assimilation

Klaus Vobig[1], Roland Potthast[1,2], and Klaus Stephan[1]

[1]Deutscher Wetterdienst, Data Assimilation Unit, Frankfurter Str. 135, 63067 Offenbach , Germany
[2]Department of Mathematics, University of Reading, Whiteknights, PO Box 220, Berkshire RG66AX, UK

**Correspondence:** Klaus Vobig (klaus.vobig@dwd.de)

**Abstract.** This paper addresses a major challenge in assimilating 3D radar reflectivity data with a Localized Ensemble Transform Kalman Filter (LETKF). In the case of observations with significant reflectivity and small or zero corresponding simulated reflectivities for all ensemble members, i.e., when the ensemble spread is vanishing, the filter ignores the observations based on its low variance estimate for the background uncertainty. For such low variance cases the LETKF is insensitive to observations and their contribution to the analysis increment is effectively zero. *Targeted covariance inflation* (TCI) has been suggested to deal with the ensemble spread deficiency (Yokota et al., 2018; Dowell and Wicker, 2009; Vobig et al., 2021). To actually make TCI work in a fully cycled convective-scale data assimilation framework, here we will introduce a *process-oriented approach* to TCI in combination with a *conditional approach* formulating *criteria* under which targeted covariance inflation is efficient.

The *process-oriented conditional* TCI addresses the challenge of underrepresented reflectivity in the prior by constructing artificial simulated reflectivities for each ensemble member based on current observations and typical convective processes. Furthermore, certain conditions are used to restrict this spread inflation process to a carefully selected minimal set of eligible observations, reducing the noise introduced into the system.

We will describe the theoretical basis of the new TCI approach. Furthermore, we will present numerical results of a case study in an operational framework, for which the TCI is applied to radar observations at each hourly assimilation step throughout a data assimilation cycle. We are able to demonstrate that the TCI is able to clearly improve the assimilation of radar reflectivities, making the system dynamically generate reflectivity that would otherwise be missing. Related to this, we are able to show that the fractional skill score of radar reflectivity forecasts over lead times of up to six hours is significantly improved by up to 10 %. All results are based on the German radar network and the model ICON-D2 covering central Europe.

## 1 Introduction

Data assimilation techniques (Lorenc et al., 2000; Kalnay, 2003; Evensen, 2009; Anderson and Moore, 2012; van Leeuwen et al., 2015; Reich and Cotter, 2015; Kleist et al., 2009; Nakamura and Potthast, 2015; Houtekamer and Zhang, 2016; Bannister, 2017) are employed for the estimation of initial conditions that are used for the initialization of dynamical forecast models. For that purpose, data assimilation techniques combine information from newly measured meteorological observations and previous model forecasts. Considering the special class of *ensemble* data assimilation techniques (Evensen, 2009, 1994; Houtekamer



and Mitchell, 1998; Evensen and van Leeuwen, 2000; Houtekamer and Mitchell, 2001; Anderson, 2001; Whitaker and Hamill, 2002; Snyder and Zhang, 2003; Houtekamer and Mitchell, 2005; Houtekamer et al., 2005; Potthast et al., 2019; Schenk et al., 2022), an ensemble of atmospheric model states is used for representing uncertainties and correlations among model variables. The usage of such an ensemble of states also allows the calculation of correlations between model variables and atmospheric observations as well as weighting the information contained in observations and model variables. Belonging to this group of

ensemble data assimilation techniques are the many versions of the particularly popular Ensemble Kalman Filter (Evensen, 2009) of which the Localized Ensemble Transform Kalman Filter (LETKF) (Hunt et al., 2007) is most relevant for this present work.

In this study, we employ the data assimilation framework KENDA (Kilometere-scale ENsemble Data Assimilation) (Schraff et al., 2016b) which combines an implementation of the LETKF that closely follows Hunt et al. (2007) and the regional

ICON-D2 model, a limited area mode configuration of the ICON (ICOsahedral Nonhydrostatic) model (Zängl et al., 2015; Prill et al., 2024) that covers central Europe. Considering the assimilation of radar data, KENDA operationally assimilates radar data by employing 3D radar observations obtained from the C-Band radar network of the German Weather service and model equivalents computed by means of the radar forward operator EMVORADO (Efficient Modular VOlume scan RADar Operator) (Zeng et al., 2016). In addition to the assimilation of 3D radar data, KENDA also includes the Latent Heat Nudging

(LHN) mechanism (Stephan et al., 2008; Schraff et al., 2016b) which is based on radar composites of radar precipitation scans.

One of the main challenges of assimilating radar reflectivities with an ensemble data assimilation system like the LETKF is dealing with observations whose corresponding background reflectivity spread is vanishing. This vanishing ensemble spread leads to an over-confidence in the background system state and, as a result, the LETKF is unable to adequately employ the information given through such observations and is effectively rejecting them—even in the presence of large discrepancies

between observed and simulated reflectivities. In practical applications it may then happen that the LETKF effectively ignores the information given through the observation of even very large reflectivity cells and is failing to synchronize the true system state, i.e. nature, with the model state.

The purpose of the TCI approach (Yokota et al., 2018; Dowell and Wicker, 2009; Vobig et al., 2021) employed in this work is to overcome the just mentioned issue, i.e., to address the issue of missing ensemble variability and, thus, make the LETKF

more sensitive to observations in cases where observations show that the ensemble is not capturing the processes adequately. To this end, artificial simulated reflectivities are constructed and assimilated. The studies Yokota et al. (2018); Dowell and Wicker (2009); Vobig et al. (2021) could show that adding spread in a targeted way can help to make the LETKF take up the observations and draw the fields into the right direction. However, it turns out that applying the scheme in a naive way to the whole domain in a cycled convective-scale framework generates a lot of noise in the system and, even though it helps in

selected situations, worsens the overall scores of both reflectivity forecasts and conventional variables.

To overcome the above problems with the TCI scheme, we will introduce two key techniques into the system. *Firstly*, the construction is accomplished by means of a specifically designed model that employs selected model variables as independent variables and has been trained on data exclusively found in the nearest spatio-temporal vicinity of early-stage convective events. This algorithm is therefore designed to capture those empirically observed correlations that are most relevant to convective





events and the involved physical processes related to their initiation. Using this algorithm for the construction of artificial simulated reflectivities and assimilating those, we expect the system to be pulled towards an overall state that is related to a (pre-)convective environment and more likely to dynamically produce reflectivity.

*Secondly*, we are employing a particular set of observation selection rules to ensure that the TCI is only applied to the most relevant observations, which usually represent only a very small percentage of the total number of observations. We found out

that these observations selection rules are essential for minimizing negative effects on the system state. Due to accumulation effects, this is particularly relevant in the context of TCI being potentially applied to all radar data at multiple time steps throughout long-term cycled data assimilation experiments.

Regarding an earlier implementation and study of the TCI approach (Vobig et al., 2021), we could already demonstrate that, in the context of non-cycled single-observation experiments assimilating only single isolated observations at single time steps,

positive effects are introduced into the system in the form of newly emerging simulated reflectivity cells. While this earlier TCI implementation is based on the same general idea, there are several substantial differences between the current version presented here and its predecessor, not only regarding methodological aspects but also regarding the type of assimilation experiments. Firstly, we completely redesigned the algorithm the TCI approach is relying on for the calculation of artificial simulated reflectivities, using pre-convective situations only. Secondly, we have established observation selection rules for

applying TCI only to carefully selected observations where a set of criteria is satisfied. Thirdly, we are processing all 3D radar observations available to our system. Lastly, we are studying longer-term NWP data assimilation cycles in an operational setup for which the TCI is applied at *each* hourly assimilation step—allowing accumulation effects to build up.

Our implementation of the LETKF in the KENDA system is described in section 2.1, the ICON-D2 model setup is summarized in section 2.2, the radar forward operator EMVORADO is explained in section 2.3, and a brief explanation of the Latent

Heat Nudging approach is given in section 2.4. We will introduce and describe the process-oriented TCI in section 3.1, the conditional approach in section 3.2, and more details on the implementation are provided in section 3.3.

The case study, upon which the numerical results presented in this work are based, and its particular setup is described in section 4.1. In section 4.2 we will demonstrate the positive effects of TCI on the basis of studies of individual cases at single times. In section 4.3 we will discuss the statistical evaluation of longer-term NWP experiments showing that the Fractional

Skill Score (FSS) (Roberts and Lean, 2008) w.r.t. the reflectivity prediction of free forecast model runs is clearly improved through the TCI by up to 10 %, while keeping the negative impact on observation error statistics at a minimum.

## 2   ICON-KENDA Ensemble Data Assimilation System

The purpose of this section is to summarize the components of our ICON-KENDA system for which process-oriented and conditional TCI has been developed. We will describe the LETKF in section 2.1, the ICON-D2 model in section 2.2, the

EMVORADO forward operator in section 2.3, and the Latent Heat Nudging mechanism in section 2.4





## 2.1 Data Assimilation: LETKF

The KENDA system (Schraff et al., 2016b) employs the Localized Ensemble Transform Kalman Filter (LETKF) as suggested by Hunt et al. (2007). This formulation allows to easily add new observations to the assimilation, while the core implementation can be kept once it is implemented. The formulation of Hunt et al. (2007), see also Nakamura and Potthast (2015, chapter 5),

solves the Kalman filter equations in ensemble space defined by the ensemble members $x^{(b,\ell)}$ for $\ell = 1, ..., L$ minus ensemble mean

$$\bar{x}^{(b)} := \frac{1}{L} \sum_{\ell=1}^{L} x^{(b,\ell)}. \tag{1}$$

We use the notation

$$\mathbf{X}^b := \left( x^{(b,1)} - \bar{x}^{(b)}, ..., x^{(b,L)} - \bar{x}^{(b)} \right) \tag{2}$$

for the matrix of ensemble differences from the mean and (for the case of linear H)

$$\mathbf{Y}^b := H\mathbf{X}^b \tag{3}$$

for the ensemble differences in observation space, $\mathbf{y}^o$ the observation vector and $\bar{\mathbf{y}}^b$ the mean of observations simulated from the ensemble. The observation error covariance matrix is denoted by $\mathbf{R}$. Now, we employ equations (20) and (21) of Hunt et al. (2007), i.e.,

$$\bar{\mathbf{w}}^a = \tilde{\mathbf{P}}^a (\mathbf{Y}^b)^T \mathbf{R}^{-1} (\mathbf{y}^o - \bar{\mathbf{y}}^b) \tag{4}$$

for calculating the mean of the analysis ensemble and $\tilde{\mathbf{P}}^a$ given by

$$\tilde{\mathbf{P}}^a = [(L-1)\mathbf{I} + (\mathbf{Y}^b)^T \mathbf{R}^{-1} \mathbf{Y}^b]^{-1}, \tag{5}$$

where we use the letter $L$ for the number of ensemble members, the notation $\bar{\mathbf{w}}^a$ for the linear coefficients of the analysis mean and $I$ for the identity matrix. $\tilde{\mathbf{P}}^a$ denotes the $L \times L$ analysis covariance in the space of ensemble coefficients. Equation (2) in

model space lead to (22) and (23) of Hunt et al. (2007),

$$\bar{\mathbf{x}}^a = \bar{\mathbf{x}}^b + \mathbf{X}^b \bar{w}^a, \tag{6}$$

$$\mathbf{P}^a = \mathbf{X}^b \tilde{\mathbf{P}}^a (\mathbf{X}^b)^T, \tag{7}$$

where $\bar{\mathbf{x}}^a$ is the analysis mean and $\mathbf{P}^a$ the analysis covariance matrix. With $\mathbf{W}$ calculated by

$$\mathbf{W} = [(L-1)\tilde{\mathbf{P}}^a]^{1/2} \tag{8}$$

as in (24) of Hunt et al. (2007) the analysis ensemble is calculated by

$$\mathbf{X}^a = \mathbf{X}^b \mathbf{W}, \tag{9}$$





where the power $1/2$ denotes the symmetric square root of the symmetric matrix $\tilde{\mathbf{P}}^a$.

It is obvious that in the case where the ensemble of simulated reflectivities has small or zero spread, the matrix $\mathbf{Y}^b$ has small or zero entries and in that case both $\mathbf{P}^a$ and the transform matrix $\mathbf{W}$ are small, such that the ensemble analysis increments given by $\mathbf{X}^a$ are small as well. The goal of targeted covariance inflation is to change $\mathbf{Y}$ in a way that the observations lead to appropriate increments in the humidity and further variables. The basic challenge of different approaches to TCI is how to construct the inflated matrix $\mathbf{Y}$ such that the increments avoid spurious noise and generated meaningful convective processes in the model propagations following the analysis steps in a cycled data assimilation framework. We will develop the process-oriented and conditional approaches in section 3.

## 2.2 NWP Model: ICON-D2

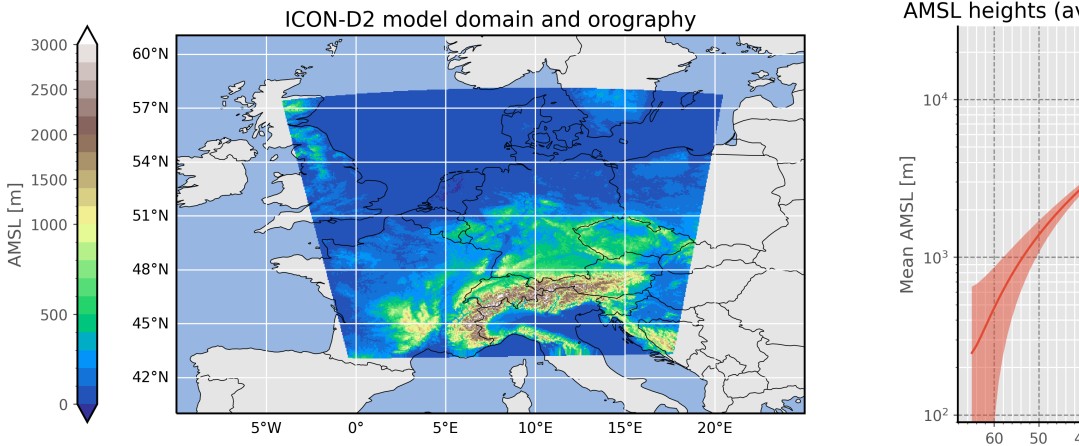

**Figure 1.** Left: Depiction of the ICON-D2 model domain covering central Europe. Colors indicate the height above mean sea level (AMSL) of the lowest ICON level that coincides with the ground level. Right: Taking the horizontal mean over the complete model domain, the mean of the AMSL heights of each ICON level is shown. Shaded areas indicate the related standard deviation.

The ICON modeling framework (Zängl et al., 2015; Prill et al., 2024) is the numerical weather prediction and climate modeling system collaboratively developed by various institutions and weather services where the Deutscher Wetterdienst (DWD) and the Max-Planck-Institut for Meteorology (MPI-M) are major contributors here. At DWD the ICON system runs operationally on a global scale, within the European subdomain known as ICON-EU, and in a convection-permitting local-area mode ICON-D2. The model domain of ICON-D2 covers all of Germany, Switzerland, Austria, and parts of the other neighboring countries, see fig. 1. Therefore, the model of ICON-D2 is very similar to that of the former operational COSMO-D2 model (Baldauf et al., 2011) which it replaced in 2021. In this work we are mainly employing the ICON-D2 configuration





for our model simulations which has a model resolution of 2.1 km, 65 vertical levels, and lateral boundary conditions provided by ICON-EU simulations.

Horizontally, the ICON model uses an unstructured triangular grid, while in the vertical dimension a distinct set of levels $\{l_i \mid 1 \geq i \geq N\}$ is defined. These levels are indexed from top to bottom, i.e., $l_1$ is the highest level far above in the atmosphere with a constant height and $l_N$ is the terrain following ground level. Furthermore, there is a continuous shift from levels of constant height to ground-following levels from top to bottom. See fig. 1 for a rough estimate of the height of each ICON level.

The ICON model solves an equation system based on a distinct set of prognostic variables. Generally speaking, a two-
component system is assumed involving dry air and water as variables where the latter may appear in all three phases. More specifically, the horizontal velocity component normal to triangle edges $v_n$, the vertical wind component $w$, the virtual potential temperature $\theta_v$, the total density of air mixture $\rho = \sum_k \rho_k$, and mass fractions $q_k = \rho_k/\rho$ are employed as variables. The subscript index $k$ refers to dry air ($k = d$), water vapor ($k = v$), cloud water ($k = c$), cloud ice ($k = i$), rain ($k = r$), snow ($k = s$), and graupel($k = g$). For a more in-depth discussion of the ICON model see Prill et al. (2024).

**2.3   Radar Forward Operator: EMVORADO**

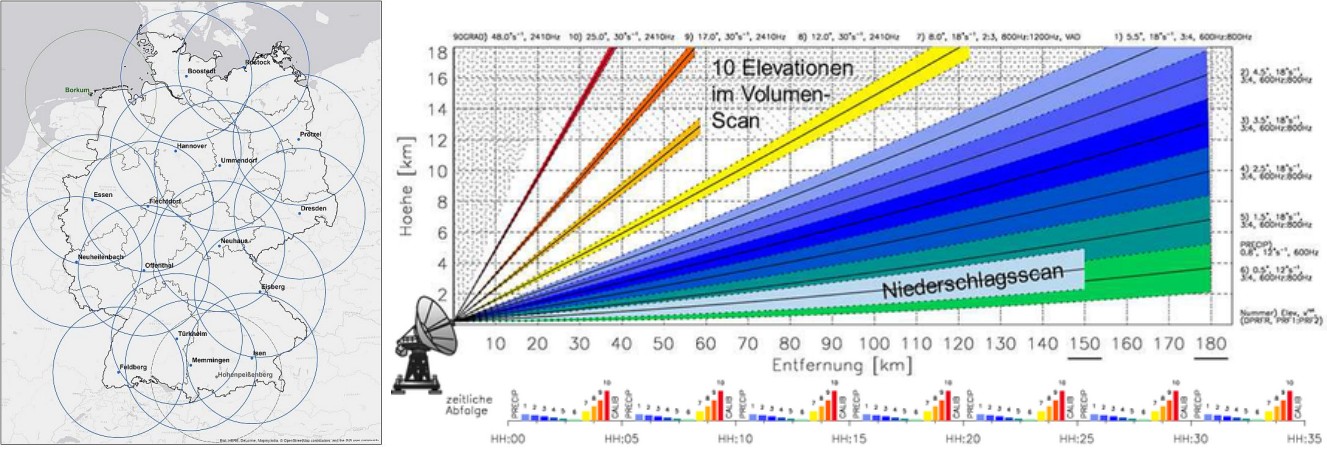

**Figure 2.** Left: Depiction of the German radar network covering the area of Germany. Each radar station is depicted as an individual circle and has a range of 150 km. Right: Scanning strategy of each radar station of the German radar network. Each of the 10 fixed elevations as well as the terrain-following precipitation scan is shown.

In the convective-scale ICON-D2 configuration of the ICON model 3D radar observations obtained from the German radar network are employed (Bick et al., 2016). The German radar network consists of 17 dual-polarization C-band Doppler radar stations that comprehensively cover Germany (see fig. 2). The scanning procedure for 3D volume scans at each radar station involves a complete $360\,°$ azimuthal sweep with a $1\,°$ resolution at ten elevation angles ranging from $0.5\,°$ to $25\,°$. Radially, the
distance reaches up to 180 km for each station with a resolution of 1 km.





To assimilate 3D radar observations, synthetic 3D radar data is derived from model variables utilizing the forward operator EMVORADO (Zeng et al., 2016) where, considering solely its single-polarization implementation in this study, Doppler velocities and reflectivities are computed. Note that simulated radar observations are produced in observation space, i.e., for each observation an associated model equivalent is computed. Furthermore, EMVORADO accounts for various intricate physical

factors related to the simulation of radar measurements like, e.g., beam bending, beam broadening, beam shielding, Doppler velocity with fall speed and reflectivity weighting, attenuated reflectivity, and detectable signal. The EMVORADO operator also allows superobbing, i.e. the local spatial averaging of observations and corresponding observation equivalents as a standard technique for assimilating spatially high-resolution observations. For more comprehensive information and specifics that are beyond the required scope of this work, please refer to Zeng et al. (2016).

## 2.4 Latent Heat Nudging

The Latent Heat Nudging (LHN) mechanism is a feature of KENDA that enables the assimilation of radar-derived precipitation rates. Note that these precipitation rates are derived from radar data supplied through the OPERA network (including the precipitation scan of the German radar network). The LHN method operates under the assumption that the precipitation rate near the surface is roughly proportional to the release of latent heat. To account for this, temperature increments are introduced

into the model. Additionally, specific humidity increments are added to ensure the conservation of relative humidity. These increments are horizontally local with respect to the underlying precipitation rates and are applied during the model run. For further details on the LHN approach and its integration into KENDA, refer to Stephan et al. (2008); Schraff et al. (2016b).

## 3 Targeted Covariance Inflation

In the following, we discuss the basic elements of the targeted covariance inflation approach (TCI) aiming for an improvement

of the LETKF assimilation of 3D radar reflectivity data. This approach is motivated by the fact that the LETKF has a fundamental deficit assimilating observations whose associated simulated ensemble spread is vanishing. In numerical applications such observations are effectively discarded by the LETKF algorithm and have no practical impact on the generated increments, which can also directly be seen through eqs. (5) to (9). The TCI approach specifically aims to resolve this issue by inflating the ensemble spread for precisely such observations.

The spread inflation is achieved by employing a specifically designed model for computing artificial simulated reflectivities for all ensemble members. The model itself is based on empirically observed correlations found in the nearest spatio-temporal vicinity of convective events. Note that the particular design of the model for the construction of artificial reflectivities is driven by the intention to make the LETKF produce additional increments that make convective initiation and the dynamic generation of reflectivity in the nearest vicinity of spread inflated observations throughout a subsequent NWP run more likely to occur.

See section 3.1 for an in-depth discussion of the construction of this model.

The overall TCI algorithm only computes and assigns artificial simulated reflectivities for observations fulfilling a certain set of conditions. As a consequence, the TCI usually only makes modifications to a very small subset of the most relevant



radar observations and the negative effects on the system state are kept at a minimum. See section 3.2 for a discussion of those conditions and a concise formulation of the overall TCI algorithm.

Finally, the TCI algorithm has to be implemented and integrated into the KENDA system for performing numerical experiments which is briefly discussed in section 3.3.

## 3.1   A Process-Oriented Regression Model for TCI

The computation of artificial simulated reflectivities $Z$ of the TCI approach is based on an application of a specifically constructed model $\mathcal{M}$. Considering the general functional form of this model, we assume a linear relationship between the simu-
lated reflectivity $Z$ (restricted to heights $h$ above mean sea level of $3000\,\mathrm{m}$ to $4000\,\mathrm{m}$) and the simulated specific humidity $q_v$ at a certain ICON level $L$. Formally, this may also be written as follows

$$\delta Z^i(\lambda, \phi, h) = \mathcal{M}\big(\delta q_v{}^i(\lambda, \phi, L)\big) = \alpha_L \cdot \delta q_v{}^i(\lambda, \phi, L), \tag{10}$$

with the ensemble member index $i$, longitude $\lambda$, and latitude $\phi$. We would like to point out that, firstly, this model is living within the ensemble perturbation space as only ensemble perturbations $\delta Z$ and $\delta q_v$ are used as variables and, secondly, that
this is a height-based approach, i.e., the algorithm only differentiates the height of radar observations and does not take the actual radar elevation angle into account.

    For determining the coefficients $\alpha_L$ we are performing a linear regression for each available value for the parameter $L$, i.e., the specific ICON level used for the independent variable $q_v$. The definite value for the parameter $L$ is then selected by finding the maximum position of the corresponding correlation coefficient $\rho_L$. As shown in fig. 3, there is a clear maximum for this
correlation coefficient of $\rho = 0.8$ at $L = 30$ (with $\alpha = 16000\,\mathrm{dBZ\,kg\,kg^{-1}}$).

    The data set employed for each of these linear regressions is constructed as follows: Initially, a raw data set is constructed which contains all relevant simulated values of an ICON-D2 assimilation cycle with 40 ensemble members running for more than two weeks[1]. Subsequently, a particular filter is applied to this raw data set such that only data representative for early-stage convective events, i.e., only spatial and temporal points in the direct vicinity of newly emerging convective cells are included.
By only including data associated with convective processes we intend to capture the most relevant correlations associated with convection and, therefore, construct a more process-oriented approach which is eventually more capable to pull the system state towards an environment that is likely to initiate convection and which helps the model to dynamically produce reflectivity.

    We would like to note at this point that it is due to internal details of the data selection process and the associated algorithm for the automatic detection of early-stage convective events that the heights of the radar reflectivities that may be inflated by
our model are, for the time being, restricted to heights of $3000\,\mathrm{m}$ to $4000\,\mathrm{m}$.

    Note that more in-depth details of the TCI model construction are beyond the scope of this work and will be presented in an upcoming separate publication.

---

[1]The setup of this particular cycle are described more in detail in section 4.1.





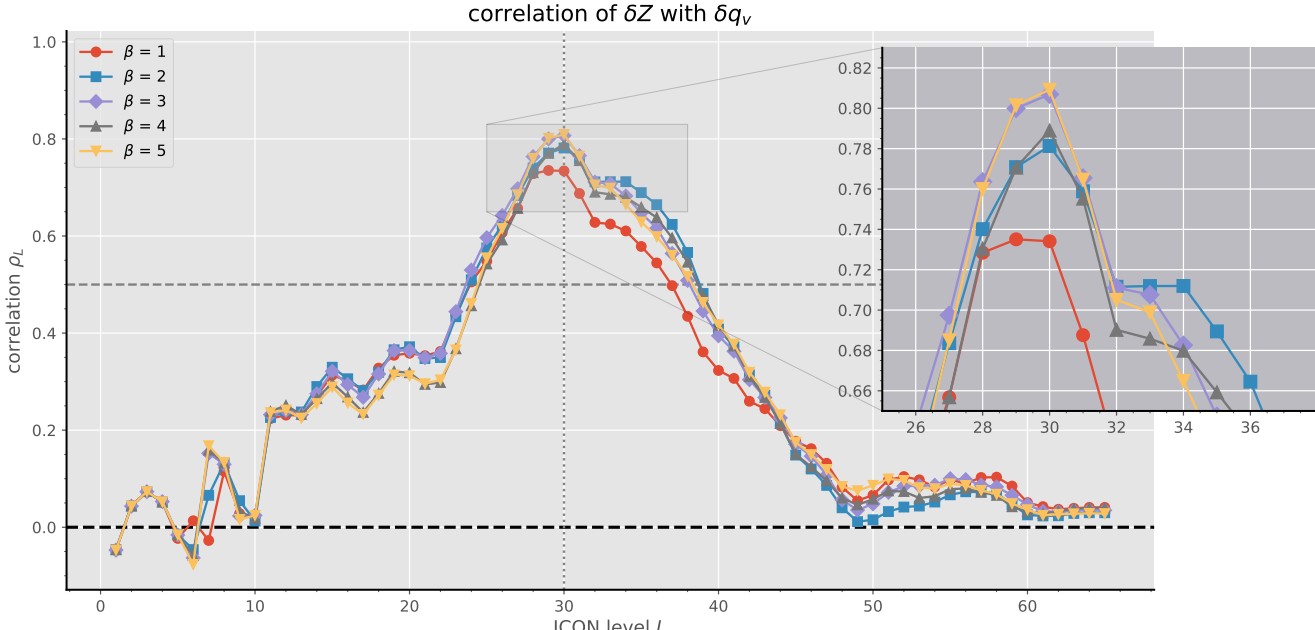

**Figure 3.** Performing a fit of eq. (10) to data that is exclusively related to early-stage convective events (see text for more information on data preparation steps), this plot depicts the resulting correlation coefficient $\rho_L$ over the ICON level $L$ of the specific humidity variable $q_v$ for each of those fits. For studying the effect of spatio-temporal displacements between reflectivities and specific humidities, a moving average with strength $\beta$ is applied to the 2D input fields for $q_v$ (see eq. (11)).

## 3.2 Conditional TCI based on Observation and Ensemble Characteristics

Another important advancement compared to earlier versions of the TCI approach (Vobig et al., 2021) is that multiple specific

conditions are required to be fulfilled before the TCI is applied for a specific observation. We found that this restriction of an application of the TCI to only a small subset of all available observations and, as a consequence, keeping the overall impact on the system state at a minimum is essential for keeping the negative effects of the TCI under control.

Some of the following operations involve the calculation of a moving average acting solely on the two horizontal dimensions. This is implemented as a centered convolution employing a normalized rectangular function of width $\beta$ (given in kilometers)

in both horizontal dimensions $\lambda, \phi$ as kernel. Denoting such a kernel as $f_\beta(\lambda, \phi)$, the processing of an arbitrary field $X$ can be formally written as

$$\widetilde{X}_\beta(\lambda, \phi, h) = \int\limits_{-\infty}^{\infty} \int\limits_{-\infty}^{\infty} \mathrm{d}\lambda' \mathrm{d}\phi' \, f_\beta(\lambda - \lambda', \phi - \phi') X(\lambda', \phi', h). \tag{11}$$

In the following, we are employing the boolean field $\mathcal{B}(\lambda, \phi, h)$ for specifying for each spatial position if the TCI should either be active (value is "true" or "1") and simulated reflectivity values should be modified, or if the TCI should be inactive





("false" or "0") and simulated values should be left unmodified. Furthermore, this field is defined as the result of a logical conjunction of several auxiliary boolean fields $\mathcal{B}_i(\lambda, \phi, h)$

$$\mathcal{B}(\lambda, \phi, h) = \prod_i \mathcal{B}_i(\lambda, \phi, h), \tag{12}$$

where each of these boolean fields $\mathcal{B}_i(\lambda, \phi, h)$ is the result of an individual condition check

**Spread-Check:**
$$\mathcal{B}_1(\lambda, \phi, h) \equiv \begin{cases} 1 & \text{if } \sigma_i[Z^{(i)}(\lambda, \phi, h)] < 0.1\,\text{dBZ} \\ 0 & \text{otherwise} \end{cases} \tag{13}$$

**Det.-Check:**
$$\mathcal{B}_2(\lambda, \phi, h) \equiv \begin{cases} 1 & \text{if } \widetilde{Z}^{\text{det}}_{\beta=10}(\lambda, \phi, h) < 1\,\text{dBZ} \\ 0 & \text{otherwise} \end{cases} \tag{14}$$

**Ens.-Mean-Check:**
$$\mathcal{B}_3(\lambda, \phi, h) \equiv \begin{cases} 1 & \text{if } \mu_i[\widetilde{Z}^{(i)}_{\beta=10}(\lambda, \phi, h)] < 1\,\text{dBZ} \\ 0 & \text{otherwise} \end{cases} \tag{15}$$

**Obs.-Check:**
$$\mathcal{B}_4(\lambda, \phi, h) \equiv \begin{cases} 1 & \text{if } Z^{\text{obs}}(\lambda, \phi, h)] > 15\,\text{dBZ} \\ 0 & \text{otherwise} \end{cases} \tag{16}$$

**Height-Check:**
$$\mathcal{B}_5(\lambda, \phi, h) \equiv \begin{cases} 1 & \text{if } 3000\,\text{m} \leq h \leq 4000\,\text{m} \\ 0 & \text{otherwise} \end{cases} \tag{17}$$

Note that we are employing $\sigma_i[X^{(i)}]$ and $\mu_i[X^{(i)}]$ for denoting the spread and mean, respectively, of a variable $X$.

With $\mathcal{B}_1$ we are ensuring that we are only making changes for observations whose associated ensemble spread is too small. The fields $\mathcal{B}_2$, $\mathcal{B}_3$, and $\mathcal{B}_4$ are taking care that the deterministic member as well as the ensemble mean have to vanish while, simultaneously, there has to be a sizeable observed reflectivity, i.e., there has to be a large discrepancy between observed and simulated values. Note that the calculation of $\mathcal{B}_2$ and $\mathcal{B}_3$ relies on simulated fields that have been pre-processed by means of a moving average (defined in eq. (11)). This is done for taking eventual spatio-temporal displacements between observed and

simulated reflectivity cells into account. Finally, $\mathcal{B}_5$ ensures that the TCI is only applied for radar observations whose height is falling into a certain height range. This is important as the previously constructed TCI model (see section 3.1) is based on observations falling into this specific height range.

    The TCI modifies the simulated reflectivity of all ensemble members employing the model $\mathcal{M}$ defined in section 3.1 but only for a specific subset of all observations which is specified by means of the logical field $\mathcal{B}$. Formally, the inflated reflectivities

$Z'^{(i)}$ of the $i$-th ensemble member are then computed via the following rule

$$Z'^{(i)}(\lambda, \phi, h) = \begin{cases} \mu_i[Z^{(i)}(\lambda, \phi, h)] + \mathcal{M}\left(\widetilde{\delta q_v}^{(i)}_{\beta=10}(\lambda, \phi, l=30)\right) & \text{if } \mathcal{B}(\lambda, \phi, h) \text{ is true} \\ Z^{(i)}(\lambda, \phi, h) & \text{otherwise} \end{cases}, \tag{18}$$





where $\lambda$, $\phi$, and $h$ loop over all discrete spatial points for which a reflectivity is measured. Note that in eq. (18) the field for $q_v$ that is entering the TCI algorithm is pre-processed by means of a moving average for taking spatio-temporal displacements into account.

Additionally, we are also modifying the observation error for each observation for which the ensemble gets inflated. Usually, we are using a global observation error of $10\,\mathrm{dBZ}$ for all radar observations in our quasi-operational setup. However, if the TCI gets applied for a specific observation $Z^{\mathrm{obs}}(\lambda, \phi, h)$ which means that $\mathcal{B}(\lambda, \phi, h)$ is true, then the observation error is reduced to $2\,\mathrm{dBZ}$. This results in much more pronounced increments and the system is significantly stronger pulled towards those observations. Note that we already saw in an earlier study (Vobig et al., 2021) that this is an advisable step for letting the 255   TCI have certain desirable effects.

### 3.3   Implementation

For integrating the TCI approach into the KENDA system (see section 2.1) the input data that is eventually entering the LETKF system is pre-processed. Usually, this input data is supplied in the form of feedback files (containing all to be assimilated superobbed[2] reflectivity observations and model equivalents) and model field files (containing, e.g., the model fields for $q_v$) 260   where there is one file per ensemble member and radar station. Processing each radar station and radar elevation separately, the TCI is implemented by performing the following steps sequentially:

1. Reading all required ICON $q_v$ model fields for all ensemble members from the model field files.

2. Reading all required observed and simulated radar reflectivity data $Z$ for all ensemble members and all radar stations from the radar feedback files.

3. Interpolation of $Z$ and $q_v$ onto a common regular two-dimensional horizontal grid (using a resolution of $2\,\mathrm{km}$).

4. Construction of the logical field $\mathcal{B}$, see eq. (12).

5. Calculation of modified simulated reflectivities for all ensemble members via the TCI algorithm, see eq. (18).

6. Writing modified simulated reflectivities back to their corresponding radar feedback files. This step involves an inverse map from the regular grid internally used by the TCI algorithm to the irregular grid internally used by the radar feedback 270     files.

7. Modification of observation errors (within the radar feedback files) for observations whose related ensemble spread got inflated.

Based on the observations and model equivalents the LETKF calculates a transform vector for the mean and a transform matrix for the ensemble perturbations (i.e. the ensemble where its mean is subtracted), which is applied to the first guess ensemble to 275   calculate the analysis ensemble. In the KENDA (Schraff et al., 2016a) implementation the calculation of the model equivalents are carried out during the model run and saved in the so-called feedback files. The calculation of the transform matrices and the execution of the transform is performed in a subsequent step by the core KENDA module.

---

[2]Superobbing refers to the process of "thinning out" radar data via spatial means.





# 4 Numerical Results

In the following, we will study the effects of an application of the TCI approach in the context of realistic NWP experiments.
In section 4.1 the details of our case study and overall setup of our experiments are discussed. The effects of the TCI on the
system state at certain single points in time are investigated in section 4.2. A statistical evaluation of the conducted longer-term
experiments is given in section 4.3 by means of observation error statistics and fractional skill scores of reflectivities.

## 4.1 Case Study Setup

For studying the effects of the TCI, we performed data assimilation cycles for ICON-D2 over the period from 2019-06-
03T00:00 to 2019-06-20T00:00 with an hourly data assimilation frequency. This specific time frame extends several days for
which individual case studies have been selected by the RealPEP research group, it includes many typical convective events.
The general meteorological situation and its temporal development over the course of the chosen time period is shown in fig. 4
by means of the spatial fraction of reflectivities above a certain threshold over time.

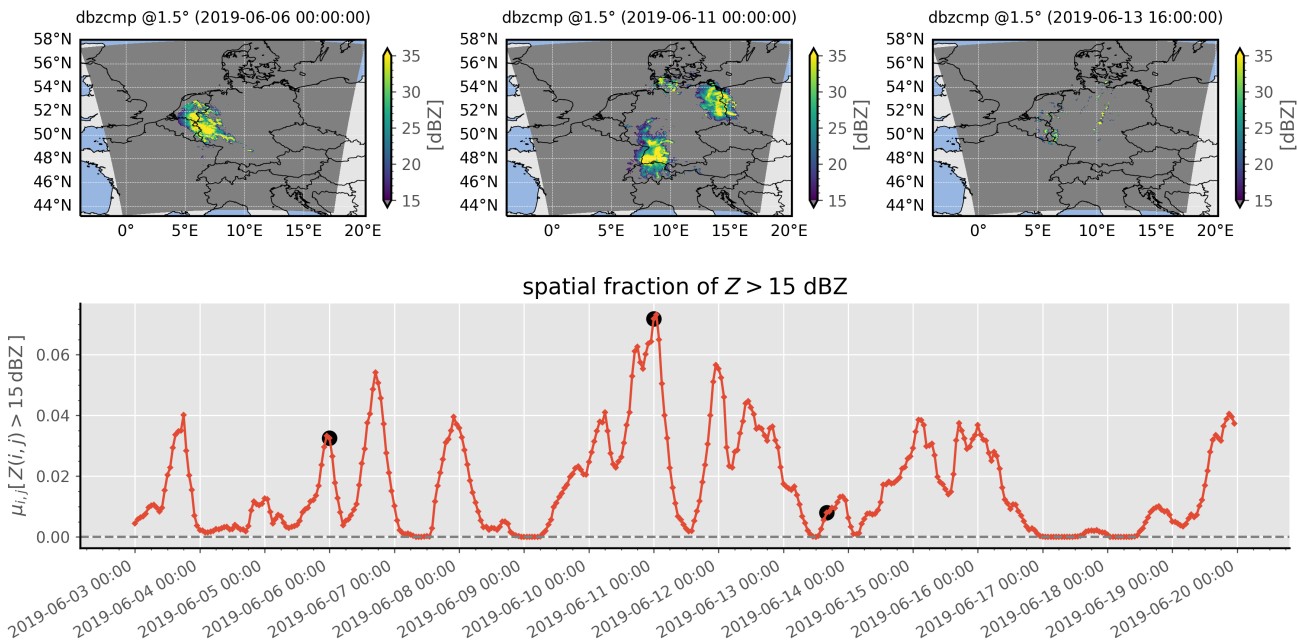

**Figure 4.** Using all available radar observations from the radar composite at 1.5 °, the lower panel depicts the fraction of all radar reflectivities
whose value is above the threshold of 15 dBZ over time. For three exemplary point in times, that are also indicated via black circles in the
lower plot, the upper three plots show the associated radar reflectivity composite.

For studying the intrinsic effects of the TCI approach, we performed two assimilation cycles: A *reference cycle* and a *TCI
cycle* which differ from each other solely by the fact that TCI is either inactive or active, respectively. The TCI cycle applies





the TCI algorithm at each assimilation step, i.e., hourly to all radar data entering the LETKF assimilation algorithm. For both assimilation cycles, we performed free forecasts every three hours with six hour lead time. During these forecasts no assimilation and, therefore, also no TCI is taking place.

Overall, the configuration of these two assimilation cycles is basically the operational configuration. This includes the as-
similation of all conventional data, the assimilation of 3D radar data from the German DWD radar network, the assimilation of radar data obtained from radar precipitation scans via the LHN mechanism, and the usage of an ensemble with 40 members. In contrast to the operational assimilation we did not include the all-sky assimilation of satellite data int our experiments, since the operationalization of these was carried out during the project execution.

At each data assimilation step the LETKF generates increments for several model variables, particularly for the temperature
$T$ and the specific humidity $q_v$, which by *incremental analysis update* (IAU) (Bloom et al., 1996) are fed into the system propagation throughout a certain time window centered around the assimilation time. However, it is important to note that for our operational setup there are no increments for other hydrometeors than $q_v$ like, e.g., $q_r, q_i, q_c$.

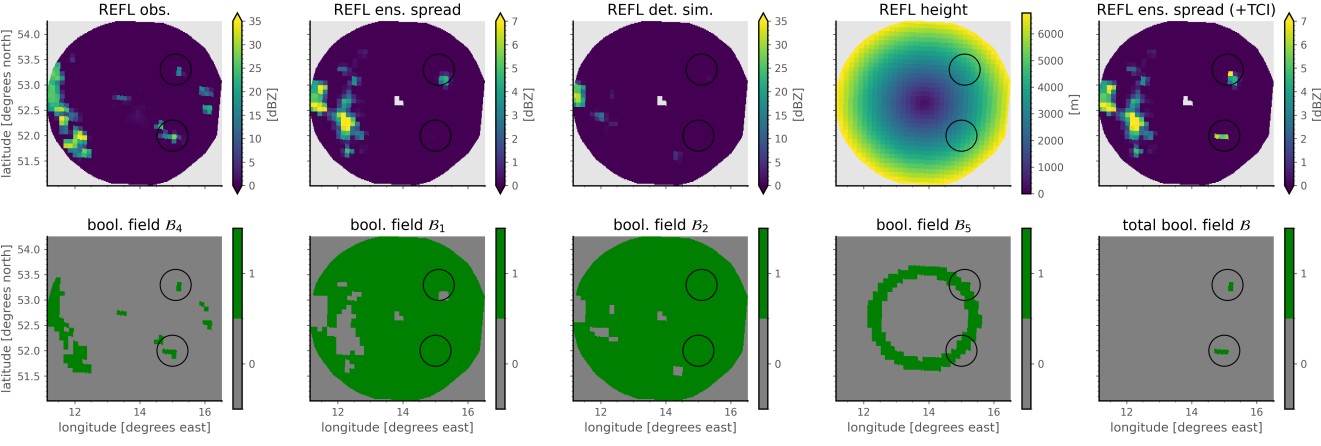

**Figure 5.** Illustration of the boolean fields $\mathcal{B}_i$ and their associated input fields, the total field $\mathcal{B}$, and the resulting inflated reflectivities (see eqs. (12) to (18)) at $t_0 = $ 2019-06-05T15:00:00. Note that solely radar reflectivity data of one single radar station (Prötzel) at one single radar elevation angle of $1.5\,^\circ$ is employed here. Each of the first four columns is related to a certain necessary condition of the TCI and depicts the result of the computation of a certain $\mathcal{B}_i$ field (bottom row) together with its related input fields (top row). The last column shows the total field $\mathcal{B}$ resulting from a logical conjunction of all $\mathcal{B}_i$ (bottom row) and the computed reflectivity ensemble spread after the TCI has been applied to all members (top row).

## 4.2 Study of Individual Cases

Let us now look more closely at the TCI effects by exemplarily studying the details of an assimilation at $t_0 = $ 2019-06-05T15:00:00
and its impact on a subsequent model run up to 1 h after $t_0$.



Let us begin with an illustration of certain internal details and immediate effects of the TCI algorithm, in particular, the construction of the total boolean field $\mathcal{B}$ and the computation of final inflated reflectivities (see section 3.2 for more information). For that purpose, fig. 5 depicts selected $\mathcal{B}_i$ fields as well as their corresponding input fields, the total boolean field $\mathcal{B}$, and the final inflated reflectivities obtained from the TCI. Note that we are only visualizing radar data of one single exemplarily chosen

radar station at a single radar elevation angle here for improving the clarity of this illustration.

Two things become apparent here: *Firstly*, there are only very few spatially connected regions for which $\mathcal{B}$ is true and, *secondly*, the TCI is successfully able to increase the reflectivity ensemble spread within these regions. Directly related to fig. 5, an aggregation (taking the mean) over all radar stations and all elevation angles of the boolean field $\mathcal{B}$ and the difference in ensemble spread is depicted in fig. 6, allowing an overview over the complete model domain. Similar to before, it becomes

evident that there are only very few spatially connected regions within the complete domain that are compatible with the imposed conditions for an application of the TCI. Looking at the depicted difference between the reflectivity ensemble spread with and without TCI, it becomes also directly clear that the TCI is able to increase the reflectivity ensemble spread if $\mathcal{B}$ is true, however, the ensemble spread is always kept unmodified for observations for which $\mathcal{B}$ is false.

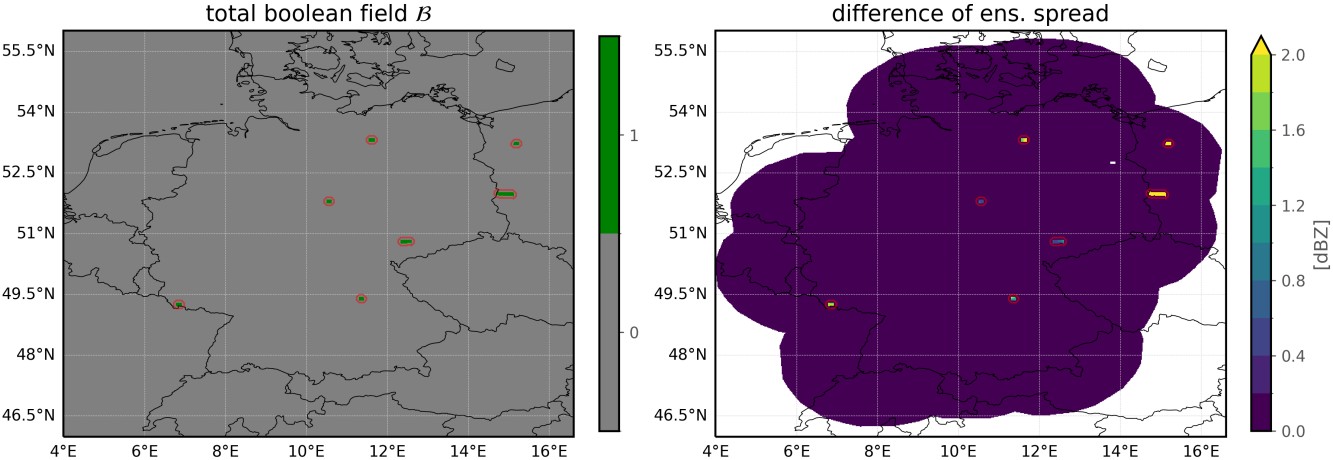

**Figure 6.** Depiction of the total boolean field $\mathcal{B}$ (first column) and the difference of the reflectivity ensemble spread with and without TCI (second column) at $t_0 = 2019\text{-}06\text{-}05\text{T}15\text{:}00\text{:}00$. While this figure is closely related to fig. 5, the 2D fields shown here are the result of an aggregation (taking the mean) over all available radar stations and radar elevation angles, allowing for an overview over the complete model domain. Red contours are used here for indicating regions for which the aggregated boolean field $\mathcal{B}$ is true, i.e., areas for which the TCI algorithm is potentially active.

As the TCI is increasing the spread by modifying all ensemble members of a only few carefully selected observations for

which the spread would be vanishing otherwise, we are enabling the LETKF to include those otherwise discarded observations. This leads to altered increments that are produced by the LETKF and we expect these increments to modify our system in a way that makes the production of reflectivity more likely. It is important to note that, firstly, reflectivity is not a prognostic but merely





a diagnostic variable and, secondly, we are not updating hydrometeor variables that are directly connected to the simulation of reflectivity (like, e.g., $q_r$). Therefore, increments do not directly affect reflectivities but the model has to dynamically respond

to increments for other variables like, e.g., temperature and specific humidity and may, eventually after a short period of time, dynamically generate reflectivity through the generation of, e.g., $q_r$ and $q_g$.

For observing how the TCI is able to let new reflectivity cells emerge, let us consider fig. 7. By considering the depicted reflectivity composite at a lead time of one hour, it becomes clear that the TCI is very often able to produce new simulated reflectivity cells that are consistent with observed reflectivities cells and that are not produced (or at least not as pronounced)

without an application of the TCI. Thus, we can already observe a positive impact of the TCI here.

At this point it is important to note that fig. 7 represents the general trend, that may be observed for other assimilation dates and lead times not shown here, very well: For regions for which the TCI is active the simulation of the TCI cycle is very often able to produce new reflectivity that are not present in the reference cycle, however, for very large observed values the corresponding simulated values of TCI cycle are usually smaller than the observed ones—which is, however, plausible as

the model merely had one hour here for dynamically responding to the additional increments introduced by the TCI and the production of reflectivity.

Furthermore, we would like to note that the source for eventual differences between the two simulated reflectivity composites may be two-fold: Firstly, the TCI has an effect through the very last assimilation step and, secondly, the TCI has also been hourly applied at many assimilations steps before the very last one at $t_0$ such that there is also an accumulation of its effects

and, therefore, a substantial divergence of the background states of the TCI and reference cycle at $t_0$.

Let us now further formalize and quantify the verification of reflectivities at a single time step as shown in fig. 7 by employing the so-called fractional skill score (FSS) (Roberts and Lean, 2008). The FSS is a popular spatial verification metric that is also used in this work for the spatial verification of reflectivities for longer-term experiments in section 4.3 and, thus, is particularly important for the overall evaluation of the TCI approach. We are denoting 2D fields for observations and associated predictions

as $Y(i,j)$ and $X(i,j)$, respectively, where $i$ and $j$ are indices for the two spatial dimensions. Considering an arbitrary 2D field $A(i,j)$, we are using the notation $\hat{A}_{\beta,\tau}(i,j)$ for referring to the so-called fraction of occurrences field. The fraction of occurrences field is defined for each spatial point as the fraction of spatial points whose value is above the threshold $\tau$ w.r.t. all spatial points lying within a certain spatial neighborhood around this point—defined through a 2D box with box length $\beta$. The FSS w.r.t. these two input fields $X$ and $Y$, the box length $\beta$, and the dBZ threshold $\tau$ may now be written as follows

$$\text{FSS}_{X,Y,\beta,\tau} = 1 - \sum_{i,j} \frac{\left(\hat{X}_{\beta,\tau}(i,j) - \hat{Y}_{\beta,\tau}(i,j)\right)^2}{\sum_{k,l}\hat{X}_{\beta,\tau}(k,l)^2 + \sum_{k,l}\hat{Y}_{\beta,\tau}(k,l)^2}. \tag{19}$$

By comparing two different predictions $X$ and $X'$ by means of the difference of their corresponding FSS with each other, we obtain

$$\text{FSS}_{X',Y,\beta,\tau} - \text{FSS}_{X,Y,\beta,\tau} = \sum_{i,j} \Delta\text{fss}_{X',X,Y,\beta,\tau}(i,j), \tag{20}$$



**Figure 7.** Visualization of radar reflectivity composites at $1.5°$ for observed values (lower left panel), simulated values of the reference assimilation cycle (upper left panel), and simulated values of the TCI assimilation cycle (upper right panel). The lead time here is $60\,\text{min}$ w.r.t. the last assimilation at $t_0 = 2019\text{-}06\text{-}05\text{T}15\text{:}00\text{:}00$. The lower right panel shows the field $\Delta\text{fss}$ computed from reflectivity data of the three other panels for $\beta = 20\,\text{km}$ and $\tau = 15\,\text{dBZ}$ (see eqs. (19) and (20)). Similar to fig. 6, red contours in all four plots are indicating regions for which the TCI was active during the last assimilation at $t_0$. Each red contour is also assigned a number for better visibility and identification and, additionally, the color of each number indicates if there is an overall positive (green) or neutral (gray) impact of the TCI on the related region.





where we inserted eq. (19), combined both sums in the resulting expression, and then implicitly defined the 2D field $\Delta\text{fss}_{X',X,Y,\beta,\tau}(i,j)$

as the argument of this combined sum. Evidently, from the sign and magnitude of $\Delta\text{fss}_{X',X,Y,\beta,\tau}(i,j)$ we may assess how much
the prediction $X'(i,j)$ improves (positive values) or worsens (negative values) the overall FSS w.r.t. the reference prediction
$X(i,j)$ and observation $Y(i,j)$.

Following these considerations, the lower right panel in fig. 7 is showing the aforementioned 2D field $\Delta\text{fss}_{X',X,Y,\beta,\tau}(i,j)$
based on simulated reflectivities of the TCI cycle for $X'$, simulated reflectivities of the reference cycle for $X$, observed reflec-

tivities for $Y$, $\beta = 20\,\text{km}$, and $\tau = 15\,\text{dBZ}$. Observing that most of the values are positive, it becomes evident that the TCI is
predominantly improving the FSS. This qualitative first impression is confirmed by computing the FSS related to the refer-
ence cycle reflectivity composite and the FSS related to TCI cycle reflectivity composite which amount to 0.796 and 0.826,
respectively. Therefore, we obtain a relative improvement of the FSS of the TCI cycle by about 3.69 %.[3]

Interestingly, fig. 7 demonstrates that the TCI not only improves the reflectivity in the near vicinity of regions for which the

TCI gets applied in the very last assimilation step (indicated via red contours) but also for many other spatial regions, hinting
at an accumulated impact of the TCI on the background state dating back to assimilations steps before the very last one.

### 4.3 Statistical Evaluation of Long-Term Experiments

Let us now proceed to a more statistical view on the TCI effects by studying different statistics and scores of longer-term NWP
experiments covering a period of about 17 days. Note that the specific configuration of these experiments, including the setup

of their assimilation cycles and free forecasts, has already been discussed in section 4.1.

#### 4.3.1 Observation Error Statistics

Figure 8 shows selected observations error statistics for the TEMP[4] relative humidity, TEMP temperature, and radar reflec-
tivities. It becomes evident that there is a slight negative impact of the TCI on the mean error of the TEMP relative humidity
w.r.t. both the analysis and the first guess especially at heights around $500\,\text{hPa}$ to $600\,\text{hPa}$ which can be interpreted as the

TCI introducing additional humidity into the simulation at those heights. However, this kind of impact is—at least to some
extent—to be expected and does not necessarily have to be regarded as a negative effect. By considering that the TCI modifies
reflectivities only within a certain height range (see section 3.2 and eq. (17)) and is employing an algorithm that is based on
correlations with the specific humidity at a certain ICON level (see section 3.1) it is plausible that—by taking cross correlations
into account—the LETKF pulls the ensemble mean towards those ensemble members with more specific humidity. Therefore,

the LETKF increases the specific humidity of the ensemble mean and the deterministic member but—taking vertical localiza-
tion into account—only within a certain height band. Considering the root-mean-square error (RMSE) of the TEMP relative
humidity, only a very small and negligible impact of the TCI becomes evident.

---

[3]Note that we are only evaluating a single point in time here.

[4]The term TEMP refers to observations obtained from radiosondes.







**Figure 8.** Observation error statistics for the TCI (label "tci-1") and the reference (label "tci-0") assimilation cycle over a period from 2019-06-03T00:00 to 2019-06-20T00:00. From top to bottom statistics for the different observation types TEMP relative humidity, TEMP temperature, and radar reflectivity are shown. From left to right the number of observations, mean error, and root mean square error (RMSE) are depicted. Note that the mean error and RMSE statistics are based on the difference of observations with their corresponding first guess values ("o-f" included in label) or analysis values ("o-a" included in label).



Similarly, the TCI has a slight effect on the mean error of the TEMP temperature which can be interpreted as the TCI increasing the temperature especially near the ground at lower altitudes. The RMSE of TEMP temperatures, however, does not

exhibit any relevant effects of the TCI.

Finally, it is strikingly—even though unsurprisingly—demonstrated that both the mean error and the RMSE of radar reflectivities are reduced through the TCI.

Overall, the largest negative impact of the TCI on observation error statistics is seen for the mean error of the TEMP relative humidity, however, the magnitude of this effect is still acceptable and for the most part to be expected. Note that a major problem

of a further advancement of an earlier version of our TCI approach as presented in Vobig et al. (2021) was—when applied to all radar and not employed within a single-observation context—a significant negative impact on observation error statistics especially on the statistics for the TEMP relative humidity. The reduction of this negative impact—while still maintaining the positive effects of the TCI—was therefore one of the main objectives that decisively guided the further advancement of the TCI towards the current version presented in this work.

### 4.3.2   Fractional Skill Score

The positive impact of the TCI is strikingly demonstrated by means of fig. 9 which depicts the fractional skill score (FSS) of radar reflectivity composites of free forecasts w.r.t. their lead time.[5] It should be noted that this verification is conducted w.r.t. the complete model domain and not only for regions for which the TCI has been active. Furthermore, the following analysis is based on full-scale data assimilation and forecasting experiments covering a period of more than two weeks and employing a

quasi-operational configuration—which especially includes an active LHN mechanism—as already discussed in section 4.1.

Let us begin our analysis with the top and middle row of fig. 9 depicting FSS statistics based on reflectivity data of all available free forecasts, i.e., there are no further restrictions on the initialization time of these forecasts. Regarding the threshold 15 dBZ, it becomes evident that the TCI is consistently improving the FSS for all depicted box lengths and lead times. It is especially striking that this positive effect is still clearly visible even after 6 h. The positive effect of the TCI on the FSS tends

to decrease with box length and the relative improvement amounts to up to 2.7 % for the box length of 1 pixel and up to 1.6 % for the largest box length of 35 pixel. Considering the plots for the threshold of 25 dBZ, a similar conclusion as before can be drawn, i.e., a positive impact of the TCI is clearly visible and amounts to up to 4.3 % for the box length of 1 pixel. Regarding the two largest thresholds of 37 dBZ and 46 dBZ the course of the FSS w.r.t. lead time becomes more erratic, however, the overall effect (averaged over lead times) of the TCI still ranges from neutral to clearly positive when taking all lead times into

account.

Furthermore, the bottom panel of fig. 9 also depicts fractional skill score improvements based solely on radar reflectivity data of model runs initialized at 12 UTC, i.e., forecasts for the time frame between 12 and 18 UTC. During these afternoon hours the positive effect of the TCI on the FSS is even more pronounced: For the threshold of 15 dBZ the FSS is improved by up to about 6 % and for the threshold of 25 dBZ there is even an improvement by up to about 10 %. As with our previous

---

[5]Note that compared to the previous usage of the FSS in section 4.2 an additional temporal aggregation of the input fraction of occurrences fields has to be carried out.





**Figure 9.** Fractional skill score (FSS) of reflectivity composites of free forecasts over lead time. The FSS calculation is based on forecasts branching off the TCI assimilation cycle (label "tci on") or the reference assimilation cycle (label "tci off"). For further details on the setup of this case study over the period of more than two weeks see section 4.1. Each panel within the upper row depicts the FSS for both experiments. Directly related to the upper row, the middle row depicts the relative FSS improvement (in percent) of the "tci on" experiment w.r.t. the "tci off" experiment. Similar to the middle row, the bottom row also depicts the relative FSS improvement but exclusively employs data from free forecasts starting at 12 UTC. For all rows the threshold used for the FSS calculation is varied column-wise and each panel shows results for several box sizes (given in pixel where 1 pixel amounts to 2.2 km).





findings, the FSS for the thresholds of 37 dBZ and 46 dBZ is rather erratic w.r.t. lead time but, when taking all lead times in account, the overall averaged effect ranges also from neutral to clearly positive.

A possible explanation for our finding that TCI clearly improves the FSS for the two lower thresholds of 15 dBZ and 25 dBZ while having a rather neutral effect for the two higher thresholds might be given through the general observation that the LHN mechanism already generates most of the large reflectivity cells and that, from a statistical point of view, our current NWP
system has the tendency to simulate too many large reflectivities but too few small ones. Additionally, TCI is only able to add reflectivity in a direct manner at each assimilation step, while a reduction of reflectivity may only be achieved via indirect accumulation effects, i.e., longer-term changes to the background system state. Therefore, it is plausible that it is much more straightforward for TCI to improve the representation of the lower reflectivity band than that of the higher one.

Overall, fig. 9 demonstrates a clear positive impact of the TCI on the FSS of radar reflectivity composites by up to 10 %.
The fact that this effect is still apparent even after 6 h hints at a more profound influence of the TCI on the background system state accumulated throughout several assimilations—which is also consistent with some of the conclusions already drawn in section 4.2.

## 5   Summary and Outlook

We have introduced and studied new process-oriented and conditional approaches to targeted covariance inflation (TCI). For
particular cases as well as for full-scale data assimilation and forecasting experiments over a period of more than two weeks we have shown that the approaches can improve the representation of convective processes in the forecasts and lead to clearly improved fractional skill scores for radar reflectivity by up to 10 %.

Details of the evaluation for different dBZ thresholds show that TCI is very good in initializing convection in the range of 15 dBZ to 25 dBZ, and has an overall neutral effect on larger precipitation cells which fall into the 37 dBZ and 46 dBZ threshold
scores. TCI as implemented through eqs. (10) and (18) currently is not dependent on the strength of the observed reflectivity, though LETKF will of course use the difference of observed and simulated reflectivity when calculating its increments.

Looking into further refinement of the scheme to further improve scores for all reflectivity bands and lead times will be a topic of future research. The sophisticated interplay of convective processes with the broader atmospheric state has the potential to be taken into account in a much deeper way. Here, machine learning techniques provide a set of very flexible non-linear
tools which can help to model more sophisticated dependencies and use them for developing a AI/ML based TCI.

The approach to construct appropriate targeted covariances in an ensemble Kalman filter is very generic and could also be employed to other types of observations. It can also be applied to other ensemble data assimilation methods such as the EnVAR (Buehner et al., 2013; Meng et al., 2019), where the observations based covariance matrix enters the scheme in the form of $HBH^T = HXX^TH^T = YY^T$ or the Localized Adaptive Particle Filter (LAPF) (Potthast et al., 2019; Schenk et al., 2022).

*Code and data availability.*   Code and data used in this work can be made available upon request to the corresponding author.





*Author contributions.* The author Klaus Vobig contributed to this paper in the following ways: Conceptualization, data curation, formal analysis, methodology, software development, validation, visualization, and writing the initial draft of this manuscript. Roland Potthast was involved in the conceptualization, funding acquisition, supervision, and review of this manuscript. Klaus Stephan contributed through conceptualization, validation, and review of this work.

*Competing interests.* The authors declare that they have no conflict of interest.

*Acknowledgements.* This work has been conducted within the context of the RealPEP project and funded by the Deutsche Forschungsgemeinschaft (DFG, German Research Foundation; 320397309).



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
