# Peer review of "On Process-Oriented Conditional Targeted Covariance Inflation (TCI) for 3D-Volume Radar Data Assimilation"

_EGUsphere, 2024_

## Referee Comment (RC2)

**Overall assessment**

The manuscript EGUsphere-2024-2876 proposes to inflate the covariance between radar reflectivity and specific humidity and reduce the declared error on radar measurements under specific conditions to ease the assimilation of observed storms that are not simulated in ensembles. The idea has merit, the work seems to have been executed competently, and the results are reasonably well described. The manuscript fails on one critical point that forces me to recommend rejection at this stage: The work, as described, is irreproducible. And because the basis of science dissemination is to provide enough information for others to reproduce the experiment described if they wish to, I judge that the manuscript cannot be published in its current form. But I believe that, with some relatively small efforts, the problem could be remediated.

**Justification and major comment**

A key to this work is the establishment of the relationship between radar reflectivity and specific humidity that was subsequently used for Targeted Covariance Inflation (TCI). How it was done remains unclear to me. The only information I could find is in L201-L204 (emphasis mine): "*Initially, a raw data set is constructed which contains all relevant simulated values of an ICON-D2 assimilation cycle with 40 ensemble members running for more than two weeks. Subsequently, a particular filter is applied to this raw data set such that only data representative for early-stage convective events, i.e., only spatial and temporal points in the direct vicinity of newly emerging convective cells are included.*". Nowhere is it explained from what or how the raw data set was constructed, what are the characteristics that make some of the simulated values "*relevant*", nor what is the "*particular filter*" in question. Alternatively, it is never clear what part of the description applies to the process used to initially select the data to establish the $\delta Z$- $\delta q_v$ relationship, and the what part applies to the algorithm used to find the pixels on which TCI was applied. If the same approach, and possibly the same data, were used for both model creation and TCI application, this was never clearly stated, even though model creation and TCI application must have been two clearly distinct and sequential steps. Yet L58, stressing that the model "*has been trained on data exclusively found in the nearest spatio-temporal vicinity of early-stage convective events*", suggests that the two are different as the grid-point selection for TCI (13)-(17) does not appear to have any temporal check. And then we have unspecified "*internal details of the data selection process and the associated algorithm for the automatic detection of early-stage convective events*" (L208-209) that seem to constrain key aspects of model creation and algorithm development. This reader is left with the impression that, intentionally or not, the authors are trying to hide key information about the research undertaken, undermining its value. And I judged that this was unacceptable in a scientific publication.

The solution, therefore, is clear: The description of how the model was established must be more thorough and less opaque, not fearing to openly admit limitations that may have made this work not as ideal as the authors would have wished.

**Other specific comments**

1) A well-known key to make convection initiation possible in a numerical model is the presence of sufficient humidity to create convectively unstable conditions. Many researchers simulating convective storms have forced high humidity in regions where radar echoes are observed (as early as Lin et al. (1993)). In many ways, your work tries to do with TCI and taking advantage of covariances what others

have done: Saturate regions with echoes to allow convective motions to occur. I believe the introduction should include some more recognition of earlier efforts that were not undertaken in a context of ensemble forecasting is the sole focus of your introduction.

Lin, Y., P.S. Ray, and K.W. Johnson, 1993: Initialization of a modeled convective storm using Doppler radar derived fields. *Mon. Wea. Rev.*, **121**, 2757–2775.

2) L56-62: What do you define as early-stage?

3) L66: Can you briefly expand what you mean by "*accumulation effects*", and/or what you fear about them?

4) L135-144: This information does not appear very relevant to the work, and I believe it could be cut. A simple reference to Prill et al. (2024) or other relevant work seems sufficient to me.

5) The same applies to the latent heat nudging section 2.4. A simple reference in the fractional skill score discussion (4.3.2) would be enough.

6) Are you assuming a linear relationship between increments $\delta Z$ and $\delta q_v$ as stated by (10) or between values of $Z$ and $q_v$ as stated in the text on L189-190? I believe it is the former.

7) Aren't you concerned about assimilating radar observations from 3-4 km altitude where radar bright bands from melting hydrometeors could affect reflectivity estimates and make reflectivity simulation more difficult, including in convective storms?

8) L279-282 seems superfluous. Consider cutting them.

9) Fig. 4 and subsequent radar images: If your algorithm uses reflectivity from 3-4 km altitude, and since you stress that the algorithm is height-based and not elevation-based (L195), why plot reflectivity at a specific elevation angle instead of at a specific height level?

10) L333-336, on explaining why the simulated reflectivities of the TCI run are smaller than the observed ones: Two other explanations could include that 1) TCI was only applied over a small height range (3-4 km), limiting the region being convectively destabilized, and 2) the sluggish 2.1-km resolution model starts its convection well after it occurred in reality and hence generally cannot evolve as quickly.

11) Having positive skill when assimilating reflectivity in convection is challenging. I was hence puzzled how you could achieve such increases in skill scores (abstract, L363, Fig. 9…) by making changes over very limited areas (e.g., Fig. 6) resulting in visually modest changes (Fig. 7). I had to relook at and fully understood the very generous FSS formula (19) to figure out why: You simply test for the fraction of pixels exceeding a reflectivity threshold in boxes of a given size to declare success. After much thinking, I decided this is fair; but I believe that you should specifically mention that verification was made (L341-342) "by employing a special version of the fractional skill score (FSS) designed by Roberts and Lean (2008) to deal with highly structured fields such as reflectivity that are particularly susceptible to double penalty (Rossa et al. 2008)" (changes underlined)

Rossa, A., P. Nurmi, and E. Ebert, 2008: Overview of methods for the verification of quantitative precipitation forecasts. In: Michaelides, S. (eds) *Precipitation: Advances in Measurement, Estimation and Prediction*. Springer, Berlin, Heidelberg. https://doi.org/10.1007/978-3-540-77655-0_16

12) If (L373-L375) "*the negative impact of the TCI on the mean error of the TEMP relative humidity w.r.t. both the analysis and the first guess [...] can be interpreted as the TCI introducing additional humidity into the simulation at those heights*", why are RH lower for the TCI experiments (top center of Fig. 8)? No attempt was made at explaining or even acknowledging the unexpected nature of this result.

13) L415-423: I believe that you are underselling the skill of your technique and being overly defensive regarding the results of Fig. 9: Why say it is neutral at 46 dBZ when it is not? Overall, the results seem as good as with the lower thresholds except that the skill evaluation is noisy because such echoes are rarer (note the very different scale of the FSS relative improvement).

**Technical corrections**

i) L51-52: "*The studies Yokota et al. (2018); Dowell and Wicker (2009); Vobig et al. (2021) could show that adding spread...*" sounds/looks funny. Do you mean "*Studies by Yokota et al. (2018), Dowell and Wicker (2009), and Vobig et al. (2021) suggest that adding spread...*"?

ii) L88-90 is a repeat of 10 lines above; consider cutting.

iii) Please improve the quality of Fig. 2, especially the right plot.

iv) (13)-(18) Radar purists will insist that the differences between two numbers in dBZ units as well as the standard deviations of reflectivities in dBZ have units of dB or of dB($Z$), the latter being more specific. This comment also applies to L251, L253, and Fig. 8.

v) L253: "*... the system is significantly stronger pulled*" should be "*the system is pulled significantly stronger*".

vi) L275: For improved clarity, add a comma between "*implementation*" and "*the calculation*".

vii, and last) The sentence on L331-L332 needs to be edited: "At this point it is important to note that fig. 7 represents the general trend, that may be observed for other assimilation dates and lead times not shown here, very well". The first underlined comma should be cut. It is also unclear to me what you are trying to say in the second underlined section; I'll let you decide how to correct it to be clearer.

---

## Author Response (AR1)

**Reply to Referee Comments**

Klaus Vobig, Roland Potthast, Klaus Stephan

March 10, 2025

**1 Introduction**

We appreciate the reviewers' thorough reading of our manuscript and their valuable feedback. Below, we address each issue by first quoting the original comment and then providing our response. Note that an updated version of our manuscript with highlighted changes is also provided as a separate document.

**2 Reply to Referee 1 (Altug Aksoy)**

**2.1 Reply to "Major Comments"**

1. "(1) On the general application of the targeted covariance inflation (TCI) method, as explained in section 3.2: (a) Can the authors provide more information about whether they have performed any sensitivity experiments when determining the individual conditional check thresholds explained through equations 13-17? It would be helpful to understand how sensitive this method is to the parameter values chosen, and thus, how sensitive the DA performance is to these choices. Please also see my detailed comment below regarding L334-336. (b) Further question/clarification is needed on the inflation of the ensemble members as explained in equation (18). I assume that there is some kind of random perturbation generated for each ensemble member as a result of the TCI algorithm. I see the mentioning of a mean and spread on L134, but I can't find these repeated anywhere else. Can you please explain this better? (c) Finally, the authors mention on L250-255 that while the default reflectivity observation error is 10 dBZ, when the TCI algorithm is applied, it is reduced to 2 dBZ so that those observations are given more weight compared to the background. I think again, it would be useful to provide more information on whether there were any sensitivity tests to come up with these values and how sensitive the DA performance was on these choices. (My detailed comment regarding L334-336 is also valid here.)"

Reply: Regarding your comment (1a), we have indeed conducted sensitivity checks on all the parameters used in the conditional check thresholds (Equations 13-17). These experiments involved varying the parameter values across a defined range and assessing the impact of these variations on the TCI application and the overall data assimilation (DA) system, as well as on short-term NWP forecasts. The results demonstrated that slight changes in the parameter values had a measurable effect on DA performance, but the method remained robust across a broad range of values. The final parameter choices were selected to strike a balance between effectiveness and noise control. We aimed to ensure that TCI was engaged when necessary to improve the DA system, but that it did not introduce excessive disruptions or noise. This approach allows the system to benefit from TCI's adaptive nature while maintaining stability in the overall DA process. A brief summary of the sensitivity checks and their results has been added to the manuscript (starting at line 275).

Regarding your comment (1b), we would like to clarify that our implementation of the Targeted Covariance Inflation (TCI) method does not introduce any random perturbations. As detailed in Equations (10) and (18), the perturbations applied to each ensemble member's reflectivity are deterministically derived from their specific humidity perturbations using a simple linear model. This approach ensures that the reflectivity perturbations are directly proportional to the specific humidity perturbations, without any random variability. We have added a note in the manuscript to further clarify this deterministic relationship (starting at line 284).

Considering your comment (1c), we previously investigated the role of observation error in our earlier TCI study (Vobig et al., 2021). While we used a different model for calculating reflectivity ensemble perturbations, we conducted similar sensitivity tests for the current TCI implementation and confirmed that the same conclusions

hold. Specifically, we found that observation error has a significant impact on TCI, as it directly controls the size of the increments applied to the ensemble. We have added a note to clarify this, outlining the results of these sensitivity checks (starting at line 291).

2. "(2) As I was reading the description of the individual cases in section 4.2, it became clear to me on L336-339 that the cycled nature of this experiment actually obscures the direct impact of the TCI method on convective initiation. It would be very helpful to generate a new experiment where the TCI method is applied in a cold-start scenario. In other words, rather than investigating the impact of TCI on a case where the "background" reflectivity is already influenced by the cycling of TCI in previous cycles, it would be more illuminating to choose a particular initial time and simply apply the TCI on the Control experiment at that particular analysis time. This simple (and cheap) analysis/forecast experiment would provide a more direct and useful picture of how the TCI method impact the generation of reflectivity in forecasts where the original ensemble had very little or no reflectivity to work with."

Reply: We have already investigated the effects of the TCI method in cold-start scenarios in (Vobig et al., 2021), where we showed that TCI is effective even when applied at a single time, generating reflectivity in cases where the ensemble background has little or no reflectivity. For the current study, we confirmed that TCI has similar effects when applied in a cold-start scenario within our experiment. Specifically, it can still generate reflectivity even when introduced at a single time, rather than through cycling. However, we chose not to include these additional cold-start experiments in this manuscript, as the primary focus is on demonstrating the effectiveness of TCI in a fully-cycled framework. Introducing cold-start experiments would shift the focus and unnecessarily lengthen the manuscript, which is already comprehensive. While we agree that cold-start experiments could offer additional insights, we believe they fall outside the scope of this study and would not significantly contribute to our main objective. However, we added a short note to our manuscript for clarifying this (see line 381 and following).

3. "(3) On the discussion of observation error statistics (section 4.3.1): It is important to caution that not all radiosonde locations should be expected to be directly influenced by the application of the TCI. After all, the TCI is applied only in a very small number of locations in each cycle, and thus it would be statistically rare for the location of radiosondes and TCI application to match. However, since TCI introduces artificially amplified humidity near the height of 4-5 km, it is plausible to expect that such effect to accumulate over time during cycling and spread out to the computational domain. But there are quantitative means to inspect this rather than speculate. For example, if this is the case, the "bias" should grow in time as more humidity bias is accumulated in the cycles. Furthermore, if indeed TCI is introducing these biases, those locations should also be associated with deficient ensemble spread in reflectivity. It would be therefore a more complete investigation if (a) time tendencies are investigated, and (b) more complete observation-space diagnostics are employed to take a closer look at bias, bias-corrected rmse (see Aksoy et al. 2009, MWR) and spread consistency ratio (see Dowell et al. 2004, MWR)."

**Reply:** We appreciate the reviewer's insightful comments on the discussion of observation error statistics. We agree that the application of TCI should not be expected

to directly influence all radiosonde locations, especially given that TCI is applied to a limited number of locations at each assimilation step and that radiosonde locations and TCI application are unlikely to align precisely.

In this context, we have already examined the evolution of humidity increments (i.e., the difference between two model runs with and without TCI). Our analysis shows how these increments drift, diffuse, and evolve across the computational domain (see also Vobig et al. 2021). However, due to the complex model dynamics, there are many non-linear effects at play, and the evolution of these increments is highly dynamic—leading, for example, to sign flips over short periods of time.

In response to your suggestion for a more complete investigation, we examined the time tendencies of observation error statistics (TEMP). Our results show that the bias near 4-5 km does not simply accumulate over time but is erratic, alternating between positive and negative values on an hourly basis. When averaged over the entire period, however, we observe a slight negative bias associated with TCI, as shown in Figure 8. Additionally, we analyzed the humidity ensemble spread and spread-skill ratio. Our findings indicate that these metrics remain practically invariant under the influence of TCI when aggregated over the entire period.

We have added a brief note to Section 4.3.1 to address the reviewer's suggestions. Beyond this addition, we believe no further investigations are necessary for the scope of this study.

4. "(4) While this manuscript focuses on the situation where the reflectivity observation suggests presence of precipitation, but the underlying model background has none, I couldn't help but wonder whether the authors are also addressing the opposite situation, i.e. spurious convection in the model background where the observations have none. Aksoy et al. (MWR 2009) and later studies have addressed this issue in various ways and I'm wondering whether the authors' DA system also addresses this "opposite" problem. From the discussion on L417-423, this doesn't appear to be the case, but a clearer explanation early on, perhaps even in the introduction, could be a useful background information for the reader to follow these results."

Reply: The opposite situation you describe—spurious convection in the model background when observations indicate no convection—is not directly addressed in our study. In our system, this issue is handled by the standard mechanisms of the LETKF data assimilation, rather than through TCI. Specifically, the problem you're referring to does not stem from a lack of ensemble spread, which is the primary concern that TCI addresses. We have added a note in the manuscript (beginning of section 3) to clarify that this opposite case is not directly tackled, particularly not by TCI.

5. "(5) The references are heavily tilted toward older publications and many books. While some of this is justifiable to introduce the basic concepts mentioned in the manuscript, the authors are strongly recommended to include relevant citations that are more recent. The issue of and research on covariance inflation, even in the context of convective forecasting and radar data assimilation, now spans nearly two decades and there are many recent publications in the literature that are very relevant to be cited. I strongly recommend that the authors go through their literature and reduce the number of citations to older manuscripts and books and instead provide a list of newer references that are relevant to the subject matter."

**Reply:** Thank you for your suggestion. We've updated the literature by incorporating recent publications and removing older sources and books. Additionally, we've added a new paragraph in the introduction discussing advancements in convective forecasting using radar data (which also contains newer references).

6. "(6) This is not "major" science-wise, but I found the common use of the present continuous tense (rather than the present simple tense) rather unnecessary and even grammatically wrong. To give an example, on L42-47, the uses "is effectively rejecting them" and "is failing to synchronize" are incorrect and should be modified as "effectively rejects them" and "fails to synchronize". This is because these are general statements and not actions occurring directly in the present time. The authors should revise their manuscript for this type of very frequent misuse."

**Reply:** Thank you for your comment. We appreciate your feedback regarding the use of the present continuous tense. We have carefully reviewed the manuscript and revised instances where the present continuous tense was incorrectly used in favor of the present simple tense, as per your suggestion.

**2.2 Reply to "Minor Comments"**

1. "L52-55: Is this issue actually documented in the references mentioned in this paragraph, or are the authors inferring this on their own? If this is documented, please provide references."

**Reply:** The issue mentioned in lines 52-55 is something we inferred on our own and is not directly documented in the references cited in this paragraph.

2. "L118-120: Can you refer to specific equation numbers so that the reader can follow this conjecture easier?"

**Reply:** Added equation numbers (see L131 and following).

3. "L120: observations -> reflectivity observations"

**Reply:** Corrected (see L135).

4. "L131: What does "COSMO" stand for?"

**Reply:** Added explanation in footnote (see page 5).

5. "L142: variables -> prognostic variables?"

**Reply:** Yes, prognostic variables. However, this whole paragraph was removed due to comments from another referee.

6. "Figure 2: Can you please translate the right panel to English? Also, the image quality of the right panel should be improved."

Reply: We translated and improved the quality of the right panel.

7. "L150: Is the 1-km radial resolution here the actual along-beam measurement resolution or some kind of average?"

**Reply:** This is an aggregated value. The actual along-beam measurements have a much higher resolution. We decided not to add this information to the manuscript as it is not directly relevant for our study.

8. "L151-152: It would be useful here to provide at least the basic forward operator equation that converts model variables to reflectivity. Is it the standard conversion that most studies utilize? (Any other factors mentioned at L155 and later obviously would not be included in this equation.)"

**Reply:** We acknowledge the suggestion to provide the basic forward operator equation. However, the equations involved are quite complex, and a full explanation would require considerable space without contributing significantly to the focus of this study. Therefore, we have decided not to include them in the manuscript.

9. "L160: Can you please be more specific about whether LHN here is something that is done outside of reflectivity assimilation?"

**Reply:** I added a note stating that both LHN and volume radar data assimilation are independent from each other (see beginning of section 2.4).

10. "L174: Eliminate the word "precisely" since you already use the word "specifically" in the same sentence."

Reply: Corrected (see L189).

11. "L176: The model itself is -> This model is"

Reply: Corrected (see L194).

12. "L181 and later: artificial -> artificially"

**Reply:** Corrected. Also corrected all other uses of "artificial" where "artificially" should have been used.

13. "L183: potential negative effects? Can you provide examples here as to what those potential negative effects could be that you're trying to minimize? Are there examples of this in the literature?"

**Reply:** We are referring to negative effects on, e.g., observation error statistics. We added a note clarifying this (see L201).

14. "L211-212: This disclaimer is actually more confusing than explanatory to me. This manuscript specifically focuses on the TCI model, so it's not clear which details of it are not relevant or beyond the scope of the present work. I suggest being either clearer about this or removing this disclaimer if it's not critical to the description of the TCI model."

**Reply:** Thank you for your comment. We have revised the entire section on data selection, filtering, and related topics in response to comments from the other referee. As part of this revision, we have removed the disclaimer you found confusing. We believe the TCI model selection/construction process is now presented more clearly.

15. "L223: for specifying -> to specify"

Reply: Corrected (see L255).

16. "Equations 14-16: Since you seem to have enough space in the left column, can you please spell out these condition checks. Specifically, it's not clear what "Det." Here refers to."

**Reply:** We spelled all condition checks of equations 14-16 out.

17. "L236: What do you mean by "deterministic member"?"

**Reply:** It seems we didn't properly introduce the concept of a deterministic member. It is used to represent the best estimate of the atmospheric state and is treated separately from the ensemble members. We added a short note explaining this at the beginning of section 2.1

18. "L286: Please indicate what "RealPEP" stands for."

**Reply:** Added that RealPEP stands for "Near-Realtime Quantitative Precipitation Estimation and Prediction" (see L323).

19. "Figure 4: Since the authors appear to prioritize the 15-dBZ observed reflectivity threshold here, I realize that it may be more helpful to change the order of the equations 13-17 so that observation check and height check (equations 16-17) are listed before the model-related checks (equations 13-15)."

**Reply:** Thank you for your suggestion. The 15 dBZ threshold in Figure 4 is not directly related to the observation check threshold. We have kept the original order of Equations 13-17, as we believe the spread check in Equation 13 is the most important and should remain prioritized in the sequence.

20. "L289 and related: cycles -> experiments? (To me at least, this would be more informative.)"

**Reply:** 'Cycle' refers to 'experiment' in this context. We have added a footnote for clarification (see page 13).

21. "L319: of only a few"

Reply: Corrected (L357).

22. "L322: It's not clear here what you mean by "production of reflectivity". Do you mean "convection initiation"?"

**Reply:** We agree that 'production of reflectivity' may be misleading in this context, as we intended to refer to the process leading to reflectivity. We have revised the text to use 'generation of reflectivity' instead (see L360).

23. "L325: Here, "like" and "e.g." are both meant to provide a list of examples. Therefore, please remove one of the two."

Reply: Removed "like" (see L362).

24. "L327: For observing -> To observe"

Reply: Corrected (see L365).

25. "L331-332: The use of the commas in the sentence makes it difficult to follow it. Please revise to make this sentence clearer."

**Reply:** Added/removed commas and split up sentences in this paragraph (see L369 and following).

26. "L334-336: Isn't this somewhat contrary to the way TCI is set up to capture convective initiation itself? When I read the description of TCI, I expected to see better examples of mature convection where otherwise the model did not produce any convection, specifically because TCI was applied during the initiation stage. The figure here shows the state of convection after one hour, which in convective time scales is a long duration to generate mature convection especially considering that typical convective overturning time scale is O(10 min). This takes me back to my major comment (1a and 1c): Could smaller reflectivity threshold values perhaps result in more robust convective initiation that is "caught" earlier in the convective lifecycle? Have you performed any sensitivity experiments along these lines? "

Reply: While we aimed for the underlying model to capture the correlations typically found during convective initiation, we cannot guarantee that TCI is applied during the initiation stage itself. The goal is to adjust the atmospheric state to make it more favorable for reflectivity generation, but the model must still dynamically adapt to these changes (we only introduce increments for specific humidity). As a result, the model's response typically takes longer than the suggested 10 minutes. Regarding sensitivity experiments, we have conducted tests to achieve more robust convective initiation, but without success. This could be due to the simplicity of our model, and we suspect that more complex models—incorporating additional variables and dependencies—might perform better in this regard. We also briefly mention in the summary that we have explored replacing the simple linear model used in TCI with more complex machine learning models.

27. "L345: spatial -> horizontal"

Reply: Corrected (see L391).

28. "L346: for referring to -> to refer to"

Reply: Corrected (see L392).

29. "L386 and elsewhere: Please avoid subjective adjectives such as "striking". What is the reason this result is striking, without any quantitative assessment? Furthermore, if this result is "unsurprising", how is it "striking" at the same time?"

**Reply:** Thank you for pointing this out. We acknowledge that the use of the word "striking" may be seen as subjective without a proper quantitative assessment. We have revised the manuscript to replace "striking" with a more objective description.

30. "Figure 9: Only looking at the row labels, it wasn't clear what was meant by "init=summed" and "init=12". If I understand correctly, these can be replaced by "init = all" and "init = 12Z", respectively, to make it much easier to follow the panels without having to refer to the figure caption."

**Reply:** We modified the labels of figure 9 as suggested.

"L412-413: Please make it clear that "afternoon" here is with respect to the local time around Germany readers from other time zones might be accustomed to associating these Zulu times with their own local times."

**Reply:** Added a short footnote stating that "afternoon" refers to local time around Germany (see page 21).

"L433: is very good in initializing -> successfully initializes"

Reply: Corrected (see L489).

31. "L434: larger -> stronger"

Reply: Corrected (see L490).

32. "L437: Double use of "further""

Reply: Removed first "further" (see L494).

**3 Reply to Referee 2 (Frederic Fabry)**

**3.1 Reply to "Justification and major comment"**

"A key to this work is the establishment of the relationship between radar reflectivity and specific humidity that was subsequently used for Targeted Covariance Inflation (TCI). How it was done remains unclear to me. The only information I could find is in L201-L204 (emphasis mine): "Initially, a raw data set is constructed which contains all relevant simulated values of an ICON-D2 assimilation cycle with 40 ensemble members running for more than two weeks. Subsequently, a particular filter is applied to this raw data set such that only data representative for early-stage convective events, i.e., only spatial and temporal points in the direct vicinity of newly emerging convective cells are included.". Nowhere is it explained from what or how the raw data set was constructed, what are the characteristics that make some of the simulated values "relevant", nor what is the "particular filter" in question. Alternatively, it is never clear what part of the description applies to the process used to initially select the data to establish the  $\delta Z$ -  $\delta qv$  relationship, and the what part applies to the algorithm used to find the pixels on which TCI was applied. If the same approach, and possibly the same data, were used for both model creation and TCI application, this was never clearly stated, even though model creation and TCI application must have been two clearly distinct and sequential steps. Yet L58, stressing that the model "has been trained on data exclusively found in the nearest spatiotemporal vicinity of early-stage convective events", suggests that the two are different as the grid-point selection for TCI (13)-(17) does not appear to have any temporal check. And then we have unspecified "internal details of the data selection process and the associated algorithm for the automatic detection of early-stage convective events" (L208-209) that seem to constrain key aspects of model creation and algorithm development. This reader is left with the impression that, intentionally or not, the authors are trying to hide key information about the research undertaken, undermining its value. And I judged that this was unacceptable in a scientific publication. The solution, therefore, is clear: The description of how the model was established must be more thorough and less opaque, not fearing to openly admit limitations that may have made this work not as ideal as the authors would have wished. "

Reply: Thank you for your detailed feedback. We have revised the manuscript to address your concerns regarding the data selection process and the relationship between the TCI model creation and its application (see section 3.1). Specifically, we have provided a more thorough explanation of how the raw data set is constructed, including the ensemble forecasts, grid resolution, and data binning process. We also clarified the filtering process used to select data representative of early-stage convective events and how this selection is robust in identifying locations associated with convective initiation. Additionally, we have made a clearer distinction between the two algorithms: the one used to construct the TCI model and the one that applies it based on specific observation selection rules, as discussed in Section 3.2. We hope these changes provide greater transparency and address the concerns raised regarding the overall methodology.

Thank you for your valuable suggestions, which have helped improve the clarity of our manuscript.

**3.2 Reply to "Other specific comments"**

1. "1) A well-known key to make convection initiation possible in a numerical model is the presence of sufficient humidity to create convectively unstable conditions. Many researchers simulating convective storms have forced high humidity in regions where radar echoes are observed (as early as Lin et al. (1993)). In many ways, your work tries to do with TCI and taking advantage of covariances what others have done: Saturate regions with echoes to allow convective motions to occur. I believe the introduction should include some more recognition of earlier efforts that were not undertaken in a context of ensemble forecasting is the sole focus of your introduction.

Lin, Y., P.S. Ray, and K.W. Johnson, 1993: Initialization of a modeled convective storm using Doppler radar derived fields. Mon. Wea. Rev., 121, 2757–2775."

**Reply:** We have expanded the introduction to acknowledge earlier efforts on how radar data has been used in improving data assimilation and NWP, particularly in enhancing convection initiation (see L33 and following).

2. "2) L56-62: What do you define as early-stage?"

**Reply:** Added a short note that we define "early-stage convective events" as regions where the model has just begun to generate significant reflectivities (see L71)

3. "3) L66: Can you briefly expand what you mean by "accumulation effects", and/or what you fear about them?"

**Reply:** We added that "accumulation effects" refer to the gradual build-up of errors or biases over time due to periodic changes to the assimilation process introduced by the TCI algorithm (see L80).

4. "4) L135-144: This information does not appear very relevant to the work, and I believe it could be cut. A simple reference to Prill et al. (2024) or other relevant work seems sufficient to me."

**Reply:** Thank you for the suggestion. We have removed most of the content, but we kept certain parts that we believe are still relevant to the context of our work (see section 2.2).

5. "5) The same applies to the latent heat nudging section 2.4. A simple reference in the fractional skill score discussion (4.3.2) would be enough."

**Reply:** We have shortened this section. However, given its importance within the ICON-KENDA framework and its relevance to interpreting our results, we believe a brief discussion is justified rather than omitting it entirely.

6. "6) Are you assuming a linear relationship between increments  $\delta Z$  and  $\delta qv$  as stated by (10) or between values of Z and qv as stated in the text on L189-190? I believe it is the former."

**Reply:** Yes, it is the former. We corrected this (see line 209).

7. "7) Aren't you concerned about assimilating radar observations from 3-4 km altitude where radar bright bands from melting hydrometeors could affect reflectivity estimates and make reflectivity simulation more difficult, including in convective storms?"

**Reply:** While radar bright bands are a general issue that can be addressed in various ways, they are not directly related to the TCI algorithm itself but rather to the broader process of assimilating radar observations. Even if an observation is affected by bright bands, resulting in an excessively large reflectivity value, the primary role of TCI is to adjust the background in reflectivity space, allowing for the assimilation of such an observation. In summary, while radar bright bands are an important consideration and are handled in different ways within our system, they are not directly tied to the TCI process itself.

8. "8) L279-282 seems superfluous. Consider cutting them."

**Reply:** We removed this part.

9. "9) Fig. 4 and subsequent radar images: If your algorithm uses reflectivity from 3-4 km altitude, and since you stress that the algorithm is height-based and not elevation-based (L195), why plot reflectivity at a specific elevation angle instead of at a specific height level?"

Reply: Thank you for your comment. You raise a valid point that we could plot reflectivity at a specific height level instead of elevation angle. However, when we say our algorithm is "height-based", we mean it simply accounts for the actual height of radar observations, while all data structures and components of our system that process radar data still use elevation angles as the primary variable. Additionally, height-based plots are less convenient, as they would require further post-processing of radar data, such as binning or interpolation (which we only performed once for constructing/selecting the TCI model). For these reasons, we chose to present the plots using elevation angle.

10. "10) L333-336, on explaining why the simulated reflectivities of the TCI run are smaller than the observed ones: Two other explanations could include that 1) TCI was only applied over a small height range (3-4 km), limiting the region being convectively destabilized, and 2) the sluggish 2.1-km resolution model starts its convection well after it occurred in reality and hence generally cannot evolve as quickly."

**Reply:** Thank you very much for your suggestions. We added them to our manuscript (see Line 369 and following).

11. "11) Having positive skill when assimilating reflectivity in convection is challenging. I was hence puzzled how you could achieve such increases in skill scores (abstract, L363, Fig. 9...) by making changes over very limited areas (e.g., Fig. 6) resulting in visually modest changes (Fig. 7). I had to relook at and fully understood the very generous FSS formula (19) to figure out why: You simply test for the fraction of pixels exceeding a reflectivity threshold in boxes of a given size to declare success. After much thinking, I decided this is fair; but I believe that you should specifically mention that verification was made (L341- 342) "by employing a special version of the fractional skill score (FSS) designed by Roberts and Lean (2008) to deal with highly structured fields such as reflectivity that are particularly susceptible to double penalty (Rossa et al. 2008)" (changes underlined)

Rossa, A., P. Nurmi, and E. Ebert, 2008: Overview of methods for the verification of quantitative precipitation forecasts. In: Michaelides, S. (eds) Precipitation: Advances in Measurement, Estimation and Prediction. Springer, Berlin, Heidelberg. https://doi.org/10.1007/978-3-540-77655-0\_16"

**Reply:** Thank you for your comment. We have modified the sentence as suggested to explicitly mention that we are using a special version of the fractional skill score (FSS) (starting at L385).

12. "12) If (L373-L375) "the negative impact of the TCI on the mean error of the TEMP relative humidity w.r.t. both the analysis and the first guess [...] can be interpreted as the TCI introducing additional humidity into the simulation at those heights", why are RH lower for the TCI experiments (top center of Fig. 8)? No attempt was made at explaining or even acknowledging the unexpected nature of this result."

**Reply:** We believe there may be a misunderstanding. The figure shows the average value of 'observation minus simulation' for RH. For both the TCI and reference experiments, this average is negative, indicating that the simulated RH values are generally larger than the observed ones. Since the magnitude of this difference is greater for the TCI experiment, it indicates that the TCI introduces additional humidity into the system, resulting in larger simulated RH values compared to the reference experiment.

13. "13) L415-423: I believe that you are underselling the skill of your technique and being overly defensive regarding the results of Fig. 9: Why say it is neutral at 46 dBZ when it is not? Overall, the results seem as good as with the lower thresholds except that the skill evaluation is noisy because such echoes are rarer (note the very different scale of the FSS relative improvement)."

**Reply:** You're right; we were somewhat defensive due to the noisiness in the results. We've revised the paragraph to present a more positive assessment of the evaluation at higher thresholds and included a note that these high reflectivities are much rarer (see section 4.3.2). Additionally, we made corresponding changes in the summary.

**3.3 Reply to "Technical Corrections"**

1. "i) L51-52: "The studies Yokota et al. (2018); Dowell and Wicker (2009); Vobig et al. (2021) could show that adding spread..." sounds/looks funny. Do you mean "Studies by Yokota et al. (2018), Dowell and Wicker (2009), and Vobig et al. (2021) suggest that adding spread..."?"

Reply: Yes, we meant that. Changed this part as suggested (see L63).

2. "ii) L88-90 is a repeat of 10 lines above; consider cutting."

**Reply:** You are correct; this is a repetition. We have removed those lines.

3. "iii) Please improve the quality of Fig. 2, especially the right plot."

**Reply:** We have improved the quality of the right plot of Figure 2 (and translated it).

4. "iv) (13)-(18) Radar purists will insist that the differences between two numbers in dBZ units as well as the standard deviations of reflectivities in dBZ have units of dB or of dB(Z), the latter being more specific. This comment also applies to L251, L253, and Fig. 8."

**Reply:** We use the unit dB(Z) now at all locations you mentioned.

5. "v) L253: "... the system is significantly stronger pulled" should be "the system is pulled significantly stronger"."

Reply: Corrected (see L291).

6. "vi) L275: For improved clarity, add a comma between "implementation" and "the calculation"."

Reply: Added a comma (see L315).

7. "vii, and last) The sentence on L331-L332 needs to be edited: "At this point it is important to note that fig. 7 represents the general trend, that may be observed for other assimilation dates and lead times not shown here, very well". The first underlined comma should be cut. It is also unclear to me what you are trying to say in the second underlined section; I'll let you decide how to correct it to be clearer."

**Reply:** We have revised this entire paragraph to clarify our meaning (see L369 and following). It should now be clearer what we intended to convey.